# Real-Time Object Classification via Dual-Pixel Measurement

**DOI:** 10.3390/s25185886

**Published:** 2025-09-20

**Authors:** Jianing Yang, Ran Chen, Yicheng Peng, Lingyun Zhang, Ting Sun, Fei Xing

**Affiliations:** 1Department of Precision Instrument, Tsinghua University, Beijing 100084, China; yangjn21@mails.tsinghua.edu.cn (J.Y.); zhangly0319@outlook.com (L.Z.); 2Department of Automation, Tsinghua University, Beijing 100084, China; r-chen21@mails.tsinghua.edu.cn; 3School of Instrument Science and Opto-Electronic Engineering, Beijing Information Science and Technology University, Beijing 100192, China; pengyicheng@bistu.edu.cn (Y.P.); sunting@bistu.edu.cn (T.S.)

**Keywords:** object classification, image-free, dual-pixel measurement

## Abstract

Achieving rapid and accurate object classification holds significant importance in various domains. However, conventional vision-based techniques suffer from several limitations, including high data redundancy and strong dependence on image quality. In this work, we present a high-speed, image-free object classification method based on dual-pixel measurement and normalized central moment invariants. Leveraging the complementary modulation capability of a digital micromirror device (DMD), the proposed system requires only five tailored binary illumination patterns to simultaneously extract geometric features and perform classification. The system can achieve a classification update rate of up to 4.44 kHz, offering significant improvements in both efficiency and accuracy compared to traditional image-based approaches. Numerical simulations verify the robustness of the method under similarity transformations—including translation, scaling, and rotation—while experimental validations further demonstrate reliable performance across diverse object types. This approach enables real-time, low-data throughput, and reconstruction-free classification, offering new potential for optical computing and edge intelligence applications.

## 1. Introduction

Object classification plays a crucial role in various fields, including remote sensing [1], autonomous navigation [2], security monitoring [3] and industrial inspection [4]. The rapid advancements in machine vision and deep learning technologies have revolutionized classification tasks, while also imposing more stringent demands on real-time performance and low power consumption. However, image-based classification techniques rely on visual data captured by imaging devices, which often contains a significant amount of irrelevant, redundant data. The pressure of high-speed data throughput has become a critical bottleneck that limits the improvement of a system’s real-time performance. Moreover, image-based methods are highly dependent on image quality, leading to poor performance in challenging conditions, such as loss of detail from low-resolution sensors, limited spectral range, and motion blur caused by high-speed movement [5].

Recent approaches such as optical computing [6,7,8,9,10] and single-pixel imaging (SPI) [11,12,13,14] have broadened new horizons for classification systems. Optical computing leverages optical elements to perform various computational tasks, delivering advantages in terms of high speed, low power consumption, and parallel data processing [7]. Qu et al. developed a super-pixel diffractive neural network for classification tasks, utilizing digital micromirror devices (DMDs) to simplify the optical system with a computational speed of 326 Hz per layer [8]. Bian et al. achieved multi-character recognition at an update rate of 100 Hz [9]. However, these approaches require the construction of multi-layer diffractive optical networks, hindering their potential for miniaturization and integration. SPI is a promising computational technique known for its wide spectral bandwidth, high sensitivity, and fast timing response [13,14]. However, as it relies on sequential time-domain illumination patterns to acquire intensity signals for reconstruction, it can be time-consuming and unsuitable for fast-moving targets.

Some researchers have conducted research on image-free classification using single-pixel detection [15,16,17,18]. By employing a spatial light modulator (SLM) for time-domain extended structured illumination to pre-code the target optical field and using a single-pixel detector to measure the modulation results, dimensionality reduction of the target’s feature information can be achieved. This enables a shift in the classification paradigm from image-centric to information-centric. A reconstruction-free classification framework was developed by Latorre-Carmona et al. [15] in 2019. Meng et al. also achieved classification by computing the moment invariants of the image [16,17]. Peng et al. present a framework for classifying fast-moving objects with shear distortion using single-pixel detection without performing image reconstruction [18]. However, these methods do not exploit the complementary properties of DMD, resulting in an excessive number of templates for optical field encoding. The polynomial moments calculated are also relatively complex, and they fail to represent other image information such as area, centroid, orientation, and ellipticity.

In this communication, we demonstrate a real-time classification method via dual-pixel measurement based on normalized central moments and the complementary nature of DMD. The proposed approach employs just five tailored illumination patterns to concurrently acquire the normalized central moments of different objects. Object classification is performed through feature recognition within the multidimensional space of moment invariants. Owing to the optimized system architecture and the rapid modulation speed of the DMD, the classification update rate can reach 4.44 kHz. A theoretical investigation was conducted to assess the method’s robustness under similarity transformations and experimental validations confirm that the system delivers accurate object classification across diverse scenarios.

## 2. Materials and Methods

The DMD serves as the core of the classification system, utilizing a micromirror array to perform high-speed binary modulation of the light field at frequencies up to 22.2 kHz. As illustrated in Figure 1, light emitted from the object is first focused through a lens (Lens 0) and directed onto the DMD via a total internal reflection (TIR) prism. Each micromirror within the array can switch between ±12° tilt angles, corresponding to ‘on’ (+12°) and ‘off’ (−12°) states. Due to the small angular separation of ±12°, the reflected and incident beams lie close to each other, which can introduce optical path conflicts and stray light interference. To overcome this, a custom-designed triple-pass TIR prism is integrated into the system [19]. In the ‘on’ state, reflected light is directed to photomultiplier tube 1 (PMT1), while in the ‘off’ state, it is sent to PMT2. The data acquisition systems (DAQ) are utilized to transmit the signal from PMTs to a computer. The binary mask of DMD ensures that the two output channels form a complementary detection mechanism. The DMD binary mask sequence for the optical path towards PMT1 is designated as mask sequence 1, while the corresponding mask sequence for the optical path towards PMT2 is referred to as mask sequence 2.

Moments and their derived functions have been widely leveraged as invariant global features for object classification [20]. Among these various types, geometric moments are the most commonly applied, primarily for characterizing the object’s shape, localization, distribution and symmetry. For a 2D image I(i,j), the (*p* + *q*)-order geometric moment *m_pq_* is given by:(1)$$m_{pq}=\overset{M}{\underset{i=1}{\sum }}\overset{N}{\underset{j=1}{\sum }}I\left(i,j\right)i^{p}j^{q}$$

To construct a mathematically invariant form under translation, the central moment of the image I(i,j) is defined as:(2)$${\bar{m}}_{pq}=\overset{M}{\underset{i=1}{\sum }}\overset{N}{\underset{j=1}{\sum }}I\left(i,j\right){\left(i-\bar{x}\right)}^{p}{\left(j-\bar{y}\right)}^{q}$$
where $\bar{x}$ and $\bar{y}$ represent the centroid of the given image, respectively, $\bar{x}=\frac{m_{10}}{m_{00}}$ and $\bar{y}=\frac{m_{01}}{m_{00}}$. It can be proved that the zeroth-order moments satisfy ${\bar{m}}_{00}=m_{00}$. In order to obtain scale invariance [21], the normalized central moment is calculated to be:(3)$$\mu_{pq}={\bar{m}}_{pq}*\frac{1}{{m_{00}}^{1+\frac{p+q}{2}}}$$

Normalized central moments provide invariance to both translation and scaling. In 1962, seven famous moment invariants to rotation were first proposed by Hu [22]. For the sake of simplicity, the first and second moment invariants were chosen for classification tasks in our method, which are:(4)$$\Phi_{1}=\mu_{20}+\mu_{02}\Phi_{2}={\left(\mu_{20}-\mu_{02}\right)}^{2}+4{\mu_{11}}^{2}$$

Given that the centroids $\bar{x}$ and $\bar{y}$ of different images varies, it is not practical to directly construct a normalized central moment template of DMD for calculation. However, normalized central moments can be simplified using the binomial theorem [23].(5)$$\mu_{pq}=\frac{\sum_{k=1}^{p}\overset{q}{\underset{l=1}{\sum }}\left(\begin{array}{c} p\\ k \end{array}\right)\left(\begin{array}{c} q\\ l \end{array}\right){\left(-\bar{x}\right)}^{p-k}{\left(-\bar{y}\right)}^{q-l}m_{kl}}{{m_{00}}^{1+\frac{p+q}{2}}}$$

The normalized central geometric moments can be directly represented as linear combinations of the target’s geometric moments, enabling efficient construction of moment invariants. This mathematical formulation provides a critical foundation for implementing DMD-based hardware acceleration in subsequent processing stages. In our previous work [24], we have developed a novel optical computing protocol that utilizes dynamically reconfigurable DMD modulation patterns to directly compute geometric moments of targets through light field manipulation, achieving 93.3% accuracy in classifying 30 different objects. However, it estimated object circularity using geometric moment templates and performed classification solely based on shape, resulting in diminished accuracy as the number of object categories grew. Here, we utilize two normalized central moment invariants mentioned above to characterize object features and perform classification within their corresponding two-dimensional feature space. This approach significantly improves classification accuracy and demonstrates robustness in more complex scenarios. The complementary dual-channel design effectively reduces the total number of required measurements. Additionally, each individual measurement inherently captures the zero-order moment of the target, enhancing the accuracy of the system. These measurements jointly encode the area, centroid, orientation, and ellipticity of objects, thereby providing a compact yet discriminative feature representation. Increasing the number of projections may introduce additional descriptors but would also lead to higher acquisition time and system complexity, which conflicts with the goal of achieving real-time classification.

According to Equation (5), the normalized central moments of an object can be computed from its lower-order geometric moments. In our previous work, we successfully employed projected DMD patterns to directly acquire the geometric moment values of the target [24]. The ideal DMD patterns $P_{pq}$ for (*p* + *q*)-order moment should be:(6)$$P_{pq}=\frac{1}{M^{p}N^{q}}\left[\begin{array}{cccc} 1^{p}1^{q} & 2^{p}1^{q} & \cdots & M^{p}N^{q}\\ 1^{p}2^{q} & 2^{p}2^{q} & \cdots & M^{p}2^{q}\\ \vdots & \vdots & \vdots & \vdots \\ 1^{p}N^{q} & 2^{p}N^{q} & \cdots & M^{p}N^{q} \end{array}\right]$$
where *M* and *N* represent the number of micromirrors in the transverse and longitudinal directions of the DMD, respectively. To convert this continuous-valued mask into a binary format, one effective approach is error diffusion dithering, which mitigates quantization artifacts by propagating the error from each pixel to its neighboring unprocessed pixels. The dithered patterns are shown in Figure 2. Therefore, the first and second moment invariants of an object can be acquired by leveraging the binary masks generated by the DMD.

## 3. Results

### 3.1. Numerical Simulation

The invariance of the proposed method under different similarity transformations is validated through numerical simulations in MATLAB R2021b (MathWorks, Natick, MA, USA), as shown in Figure 3. The scaling and rotational invariance of the DMD-based object classification method was first characterized. To demonstrate the generality of the method, we selected two different objects (a moon and an airplane) as test subjects. In the simulation of scale invariance, we gradually reduced the size of the initial objects to 60% of their original size, with a scaling factor decreasing by 0.01 at each step, resulting in 41 measurement values in total (Figure 3a). By incorporating complementary dual-pixel detection, we computed and plotted the corresponding invariants 1 and 2 with respect to the scaling factor (Figure 3b).

In the simulation of rotation invariance, each object was rotated around its centroid, starting from 0° and gradually increasing the angle to 360° in 10° intervals, while the corresponding invariants were calculated at each rotation angle (Figure 3c). It can be observed that invariant 1 remains almost unchanged with rotation. Due to image digitization and the binarization effect of our method [24], invariant 2 exhibits some fluctuation with respect to the rotation angle. In addition, it should be noted that the error-diffusion dithering algorithm we currently employ propagates quantization errors along the horizontal or vertical directions. As a result, the error transmission is not strictly isotropic, which constitutes an inherent systematic error of our method. However, by considering both invariants together, accurate classification results can still be obtained. The details of this process will be described in the following experimental section.

To further demonstrate the effectiveness of our method, we translated the object within a specified region of the image plane and calculated the error between the theoretical values of the two invariants and the values obtained using our method. The simulation results (Figure 4) demonstrate that the proposed method maintains strong performance under translational transformation. And the RMSE of invariant 1 and invariant 2 are calculated to be 0.0230 and 0.0027, respectively.

### 3.2. Experimental Validation

Building upon the theoretical framework and simulations presented above, we have constructed an experimental system (Figure 5) to further evaluate the effectiveness of the proposed method. The components of the experimental system and their respective functions are outlined as follows: A DLP technology-driven dynamic object simulator is connected to computer 1 for real-time simulation of various objects. Lens 0 focuses the light emitted by the target simulator onto the front surface of the DMD (DLP7000, Texas Instruments, Dallas, TX, USA), while the total internal reflection (TIR) prism is utilized to fold the optical path and compress the system’s spatial layout, facilitating dual-pixel detection. The DMD generates structured mask sequences in real time, which, when applied to the incident light field, enable the computation of the object’s normalized central moments of various orders. The two optical paths exiting the TIR prism are symmetrical. Mirrors 1 and 2 are employed to redirect the optical path, ensuring a compact system. The outgoing light is then converged by lenses 1 and 2 onto the PMTs (PMT1001/M, Thorlabs, Newton, NJ, USA) for detection, and the signals are transmitted to the computer via a data acquisition card for real-time analysis and object classification.

To demonstrate the versatility of our method and further validate its invariance to rotation, translation, and scaling, we selected five distinct objects from the MPEG-7 dataset [25] and applied various similarity transformations randomly. Using the dynamic target simulator, we generated images corresponding to different objects, which were then modulated by the DMD. The signals collected by the PMT were subsequently used for classification. Since the numerical values of the moment invariants are relatively small, a logarithmic transformation was applied to facilitate visualization. The transformed values $\phi_{1}$ and $\phi_{2}$ were then used as the basis for classification, where α is the weighting coefficient:(7)$$\phi_{1}=\alpha \cdot \left|\log_{10}\Phi_{1}\right|\;\phi_{2}=\left|\log_{10}\Phi_{2}\right|$$

As shown in Figure 6, five object categories were classified using the proposed method. Since the maximum modulation rate of the DMD is 22.2 kHz and each classification requires the projection of five binary patterns, the system achieves a real-time classification rate of 4.44 kHz. The coefficient α is selected to be 5. Furthermore, Figure 6b illustrates that relying on a single normalized central moment invariant as the classification descriptor is insufficient for certain object pairs. For instance, when only invariant 1 is used, the lmfish and deer are difficult to distinguish; conversely, using only invariant 2 makes it challenging to differentiate the apple from the crown. By constructing a two-dimensional feature space using both invariants, the system not only maintains high-speed real-time classification but also significantly improves classification accuracy.

To quantitatively assess the proposed method, we used the first column in Figure 6a as the reference objects (numbered as 0), providing standard moment invariant values. The remaining four columns, obtained through similarity transformations, serve as test samples. For each object, we calculated its distance from all five reference categories in the moment invariant space. We computed the average distance between each of the four test objects within the same category and the standard objects from different categories. The results are summarized in Table 1. Experimental results demonstrate the robustness of our classification method under various similarity transformations, showing clear differentiation among multiple object categories. Part of the experimental error originates from the fact that the proposed method is not perfectly isotropic, as already discussed in Section 3.1. In practical applications, our method involves a pre-calibration step, where representative invariant values are computed in advance for each known object category. Classification can be achieved by calculating the moment invariants of a given object and measuring its two-dimensional distance from the reference values of standard categories. If this distance falls below a predefined threshold derived from pre-calibration, the object is considered to belong to the corresponding category. In this experiment, the threshold is chosen to be 0.1~0.15. To further demonstrate the applicability of our method on larger test sets and to provide a performance comparison with previous approaches. We utilize all 70 object categories from the MPEG-7 dataset to serve as the experimental test set. The results are shown in Table 2. We calculate the moment invariants of the different objects for classification, and the results show that our method maintains an accuracy of over 80% even when the number of target object categories increases to 70.

## 4. Discussion

The proposed dual-pixel classification system addresses key limitations of conventional image-based methods by eliminating the need for full image acquisition and reconstruction. By encoding object features directly through structured binary DMD masks, the method significantly reduces data redundancy and computation time.

However, the study also reveals some challenges. Although invariant 1 is highly stable across transformations, invariant 2 exhibits minor fluctuations under rotation, likely due to binarization artifacts. Nevertheless, combining both invariants ensures reliable object differentiation. Moreover, the proposed method is currently applicable only to single-object classification. In scenarios involving multiple objects, a preliminary segmentation step is required to isolate individual targets before projecting the corresponding masks for information acquisition. Further improvements could be achieved by designing a multi-object classification method and incorporating more moments and their invariants to enhance classification separability in higher-dimensional spaces.

## 5. Conclusions

In this communication, we have developed a real-time, image-free object classification method that leverages dual-pixel detection and DMD-based structured illumination. By extracting low-order normalized central moment invariants through optical modulation, the system enables fast and accurate classification across varying object types and transformations. Simulation and experimental results confirm that the method achieves high robustness and efficiency with an update rate of up to 4.44 kHz. The compact, complementary detection architecture and elimination of image reconstruction offer a promising foundation for future integration into miniaturized optical computing systems. Future work may involve learning-based enhancements to enable more complex recognition tasks across a broader range of applications.

## Figures and Tables

**Figure 1 sensors-25-05886-f001:**
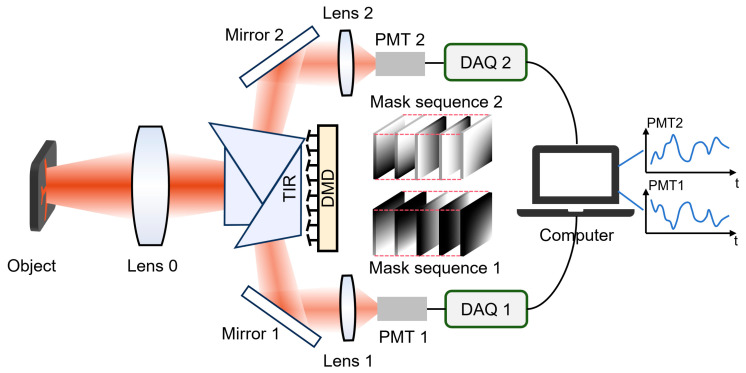
Schematic diagram of the real-time object classification system.

**Figure 2 sensors-25-05886-f002:**
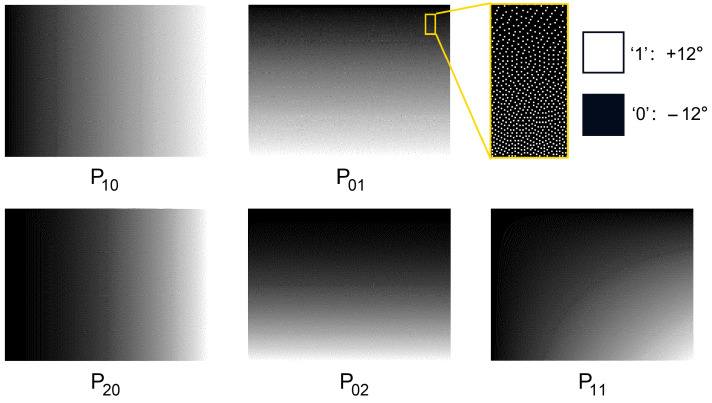
DMD binary mask for calculating the object’s geometric moment.

**Figure 3 sensors-25-05886-f003:**
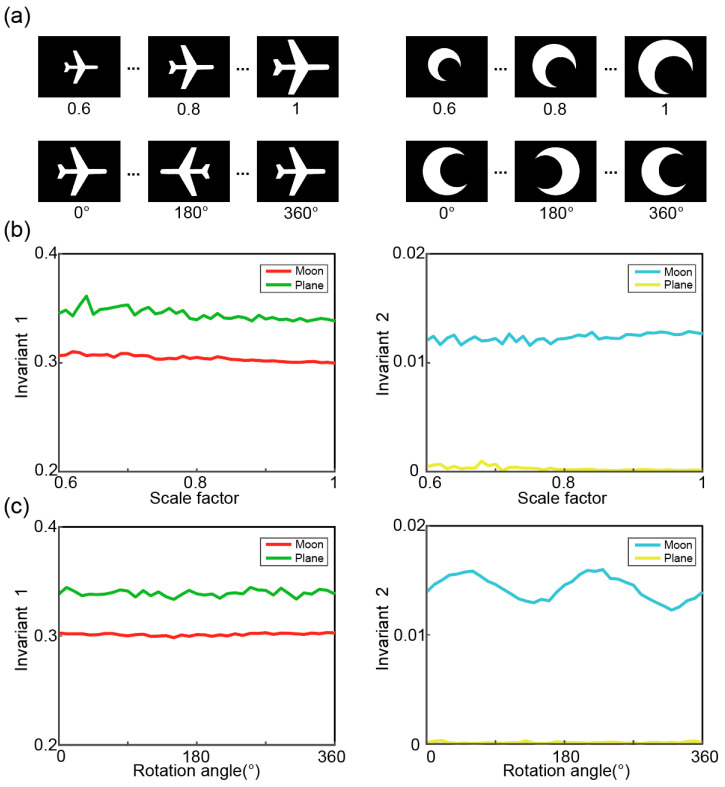
The scaling and rotational invariance of the proposed method. (**a**) The test object. (**b**) The calculated results of Invariants 1 and 2 under the scaling transformation. (**c**) The calculated results of Invariants 1 and 2 under rotational transformation.

**Figure 4 sensors-25-05886-f004:**
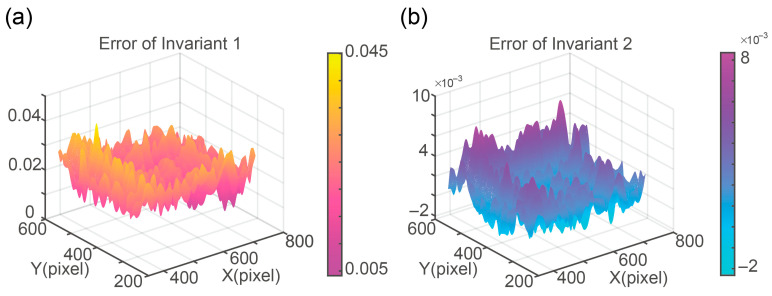
Visualization of Invariant 1 and 2 errors across the DMD plane. (**a**) Error of invariant 1. (**b**) Error of invariant 2.

**Figure 5 sensors-25-05886-f005:**
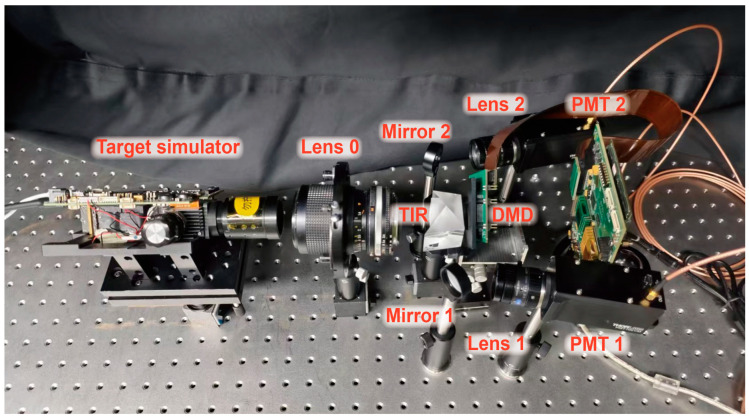
Experimental setup for real-time object classification.

**Figure 6 sensors-25-05886-f006:**
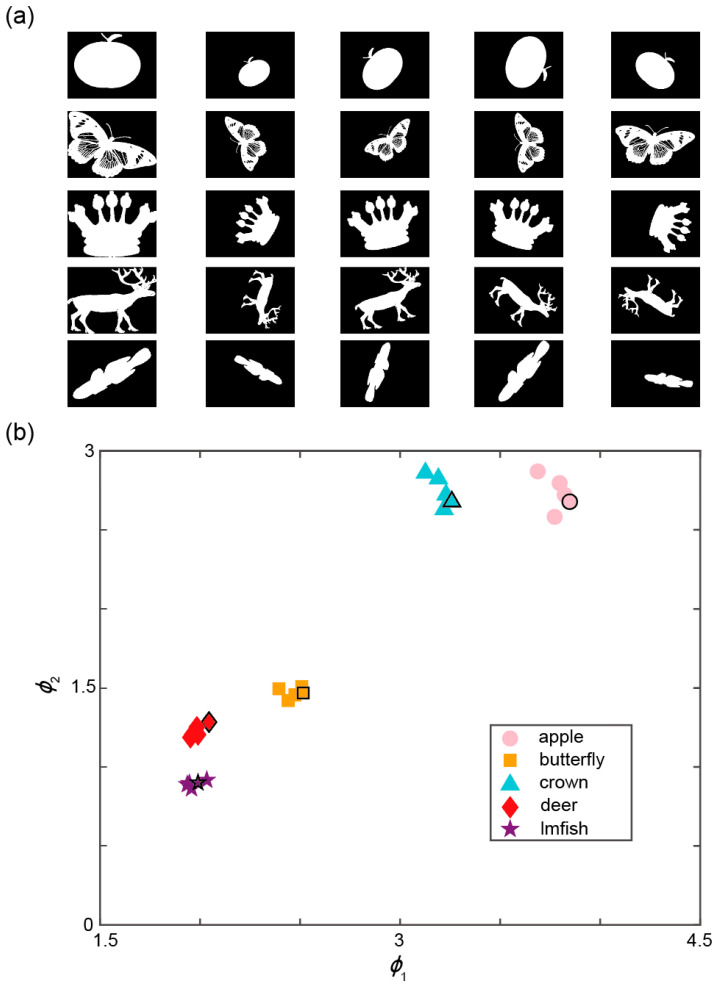
Experimental validation of system classification performance. (**a**) Different test objects displayed on the dynamic target simulator. (**b**) The measured invariant values $\phi_{1}$ and $\phi_{2}$ of different test objects. The referenced standard objects in each category are distinguished by bold black bounding boxes.

**Table 1 sensors-25-05886-t001:** Average distance between test objects and standard objects.

Object Type	Average Distance				
	Apple0	Butterfly0	Crown0	Deer0	Lmfish0
Apple	0.1377	1.7904	0.5275	2.2612	2.5639
Butterfly	1.8443	0.0741	1.4515	0.4512	0.7365
Crown	0.6666	1.4585	0.1251	1.8700	2.2151
Deer	2.3799	0.6006	1.9496	0.1027	0.3141
Lmfish	2.5929	0.7953	2.2035	0.4015	0.0484

**Table 2 sensors-25-05886-t002:** Classification results of different objects in comparison with previous method.

Object Type	Classification Accuracy	
	Our Method	Previous Method [24]
30	96.7%	93.3%
50	90%	82%
70	81.4%	68.6%

## Data Availability

The data presented in this study are available from the corresponding author upon reasonable request.

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
