# Peer review of "Real-Time Object Classification via Dual-Pixel Measurement"

_sensors, 2025, doi:10.3390/s25185886_

Round 1

Reviewer 1 Report

Comments and Suggestions for Authors

In this paper, the authors present a high-speed and image-free object classification method based on dual-pixel measurement in single-pixel imaging. The five illumination patterns are designed based on geometric moments. The reviewer thinks this work makes a weak case for novelty and significance. The reasons are given below. First, the idea of designing illumination patterns based on geometric moments has been attempted in many previous works for fast object localization in single-pixel imaging. It is not proposed for the first time in this paper. Second, the five illumination patterns can be used to determine the centroid position of the object in a simple way. But they may not provide adequate information for object classifiction. Third, only some simple binary object examples are used to test the proposed method in this paper. More complicated image datasets shall be attempted. Fourth, the designed illumination patterns are fixed for different classifiction tasks in this work. It is favorable that the optimized illumination patterns are adaptively changed for classifying different kinds of object images. Overall, the reviewer does not recommend the acceptance of this paper.

Reviewer 2 Report

Comments and Suggestions for Authors

This paper presents an interesting method for real-time object classification using DMD. The problem is well introduced together with the proposed solution. 

The data results of the simulations are presented well, but a question stays open about the reason of the variation of φ2 with respect to rotation. The author should describe and maybe demonstrate why the reason of the variation is associated to digitization. Citations are reported, but a brief explanation and proof could help the reader.

The same variation could be the case of the large spread of the vertical axis in figure 6. The authors could comment on this. 

In addition, the average value of the distance of between measured points and reference images is used to demonstrate the feasibility of the classification. It is not clear how the suggested threshold (between 0.1 and 0.15) was defined. The results could also include the variance of a larger set of measurement to support the thesis, which seems reasonable from this communication. If a larger test set is not available, at least the distribution of the available ones should be described (variance of distances).

The average distance between test objects and reference could be biased by the choice of the reference. The authors should highlight in figure 6 which of the points correspond to the reference images. 

Reviewer 3 Report

Comments and Suggestions for Authors

This paper presents a novel optical framework for object classification based on dual-pixel detection and normalized central moment invariants using a digital micromirror device (DMD). The authors claim that only five binary illumination patterns are sufficient to achieve classification at a high update rate of 4.44 kHz. Both simulations and experimental validations are provided, demonstrating robustness under similarity transformations (translation, scaling, rotation).

Some points should be considered:

(1) Threshold Selection

A key weakness of the paper is its reliance on an empirical distance threshold (0.1–0.15) for classification. As described in the Results section (lines 226–229, p.9), an object is considered to belong to a reference category if its distance in the moment-invariant space falls below this range. However, this threshold is derived solely from experiments involving five object categories (Section 3.2, lines 195–204, p.7). It is unclear whether the same threshold would remain effective if the dataset were expanded to 10, 20, or more categories. Without a specific method for adaptively adjusting thresholds, the generalization of the method on larger-scale classification tasks remains uncertain.

(2) Number of Binary Projections

The authors claim that only five binary projections are required for classification (Abstract, lines 17–19, p.1; Results, lines 204–206, p.7), but the paper does not provide a rigorous theoretical justification for this choice. While it has been experimentally shown that five tailored patterns suffice for the selected categories, it remains unclear whether increasing the number of projections (e.g., to six or more) might further improve classification robustness or accuracy. A comparative experiment or at least a theoretical analysis would be necessary to establish why five projections is optimal.

(3) Limited Evaluation

The experimental validation is limited to only 5 out of the 70 categories in the MPEG-7 shape dataset (Sec. 3.2, lines 195–197, p.7), which makes it difficult to assess the generalization capability of the proposed approach. To substantiate robustness, additional results on larger subsets (e.g., 10 and 20 randomly selected categories) should be included. Furthermore, the paper does not provide quantitative comparisons with existing SOTA single-pixel classification approaches.

Round 2

Reviewer 1 Report

Comments and Suggestions for Authors

The reviewer's comments have been well addressed. This paper can be accepted for publication. As one minor revision, in the introduction part, for fast object classification in single-pixel imaging, in addition to the works stated, the following early works [r1,r2] may be included as well.

[r1]S. Jiao, J. Feng, Y. Gao, T. Lei, Z. Xie, and X. Yuan, "Optical machine learning with incoherent light and a single-pixel detector," Opt. Lett. 44(21), 5186-5189 (2019)

[r2]P. Latorre-Carmona, V. J. Traver, J. S. Sánchez, E. Tajahuerce, Online reconstruction-free single-pixel image classification, Image and Vision Computing 86, 28-37 (2019)

Author Response

We sincerely thank the reviewer for the positive evaluation and for the helpful suggestion. In the revised manuscript, we have added the suggested references [r1, r2] in the Introduction to provide a more complete background on early single-pixel classification works. We note that reference [r2] (Latorre-Carmona et al., 2019) was already included in our original manuscript as Ref. [16].